# Electrocatalysis for Oxygen Reduction Reaction on EDTAFeNa and Melamine co-Derived Self-Supported Fe-N-C Materials

**Mengfan Shen [1,2], Ziwei Meng [2], Tong Xue [2,3], Hongfang Shen [2] and Xiang-Hui Yan [1,2,*]**

[1] Key Laboratory of Powder Material & Advance Ceramics, North Minzu University, Yinchuan 750021, China; smf15197845460@163.com

[2] School of Materials Science and Engineering, North Minzu University, Yinchuan 750021, China; mzw13895397283@163.com (Z.M.); tong_xue@nun.edu.cn (T.X.); shen_hongfang@nun.edu.cn (H.S.)

[3] International Scientific & Technological Cooperation Base of Industrial Waste Recycling and Advanced Materials, North Minzu University, Yinchuan 750021, China

[*] Correspondence: 2014058@nun.edu.cn and yanxianghui@tsinghua.org.cn; Tel.: +86-951-2067378

**Abstract:** To explore high-performing alternatives to platinum-based catalysts is highly desirable for lowering costs and thus promoting fuel cell commercialization. Herein, self-supported Fe-N-C materials were prepared by the pyrolysis of dual precursors including EDTA ferric sodium (EDTAFeNa) and melamine (MA), followed by acid-leaching and final annealing. Towards an oxygen reduction reaction (ORR) in 0.1 M KOH, the as-prepared MA/EDTAFeNa-HT2 delivered onset ($E_{onset}$) and half-wave ($E_{1/2}$) potentials of 0.97 and 0.84 V vs. RHE, respectively, identical with that of a state-of-the-art Pt/C catalyst, accompanied with predominantly a four-electron pathway. The introduction of MA and extension of acid-leaching promoted a positive shift of 50 mV for $E_{1/2}$ relative to that of only the EDTAFeNa-derived counterpart. It was revealed that the enhancement of ORR activity is attributed to a decrease in magnetic Fe species and increase in pyridinic/quanternary nitrogen content whilst nearly excluding effects of the graphitization degree, variety of crystalline iron species, and mesoscopic structure. The usage of dual precursors exhibited great potential for the large-scale production of inexpensive and efficient Fe-N-C materials.

**Keywords:** self-supported Fe-N-C materials; dual precursors; EDTA ferric sodium; melamine; oxygen reduction reaction



## 1. Introduction

Owing to a higher over-potential and more complex reaction mechanism for the cathodic oxygen reduction reaction (ORR) than those of the anodic hydrogen oxidation in $H_2$-$O_2$ fuel cells during electrocatalysis, the energy conversion efficiency thus primarily depends on its ORR catalysis [1–4]. To pursuit highly efficient Fe-N-C catalysts as alternatives for replacing the precious and scarce platinum-based catalysts has been a promising strategy to reduce the cost and promote the large-scale development of fuel cells [5–7]. Among various synthetic methods involved in self-supported Fe-N-C catalysts (SSFNCCs), a template-free strategy has gained increasing attention due to fewer preparation procedures without the addition of a soft template or removal of a hard template [8–11].

By pyrolyzing a ferrous EDTA chelate, syntheses of SSFNCCs have been reported [12–14]; however, their electrocatalytic performance is still inferior to that of the Pt/C catalyst. These results imply that it is inadequate and even difficult for a single nitrogen-containing precursor to concurrently control the porous structure, nitrogen content, and coordination of Fe atoms with N and C, which are key factors affecting ORR performance [15–19]. To date, pyrolysis of dual precursors could serve as a promising route to optimize structures and properties of the resultant SSFNCCs, thus tailoring the ORR performance [20–24]. Lou et al. [20] reported porous N-doped carbon matrix-supported N-doped carbon nanotube assemblies with embedded iron carbide nanoparticles ($Fe_3C$@N-CNT assemblies), which

were fabricated via a dual-MOFs pyrolysis route where Fe-based MIL-88B nanorods were firstly confined in a Zn-based ZIF-8 polyhedron host. The as-obtained $Fe_3C@N$-CNT assemblies was bestowed with well-defined overall morphology, mesoporous structure, and highly dispersed small-sized $Fe_3C$ nano-crystallites, which contributed to the higher ORR activity as compared with the Pt/C catalyst ($E_{1/2}$ of 0.85 and 0.83 V, respectively). Direct Ar-pyrolysis from a dual nitrogen precursor of poly-o-phenylenediamine (PoPD) and melamine (Mela) with $FeCl_3 \cdot 6H_2O$ was investigated for the preparation of Fe/N co-doped carbon materials (denoted as Fe/oPD-Mela) [25], which possessed a high surface area and high content of active nitrogen species such as graphitic and pyridinic nitrogens. The as-prepared Fe/oPD-Mela exhibited a four-electron pathway with an impressive electrocatalytic ORR activity in terms of $E_{1/2}$, outperforming and approaching that of state-of-the-art Pt/C in alkaline and more challenging acidic electrolytes, respectively. Although dramatic advancements have been achieved, great efforts remain to be made for further improving ORR performance, clarifying the nature of components in catalysts and corresponding mechanisms, and facilitating the practical applications of these SSFNCCs in fuel cells.

Herein, low-cost EDTAFeNa containing C, Fe, and N atoms and melamine (MA) were employed as a dual precursor to fabricate SSFNCCs via facile pyrolysis and acid-leaching for ORR in 0.1 M KOH. As we previously reported, SSFNCCs generated from the pyrolysis of sole EDTAFeNa showed a low nitrogen content and large-sized crystalline Fe particles, which could be responsible for relatively low ORR activity due to the low density of $CN_x$ and $FeN_xC_y$ moieties involving N species. Although active sites in Fe-N-C materials are still not clear, the $CN_x$ and $FeN_xC_y$ moieties have been proposed as active sites [17,26–28]. To potentially increase the numbers of above-mentioned active sites, N-rich MA was proposed to proportionally combine with EDTAFeNa to improve the percentage of nitrogen species and dispersion of Fe species during pyrolysis. In addition, the acid-leaching time and pyrolysis temperature were investigated as well. The resultant SSFNCCs (denoted as MA/EDTAFeNa-HT2) displayed excellent ORR activity, reflected by the identical $E_{onset}$ and $E_{1/2}$ to the Pt/C catalyst (0.97 and 0.84 V, respectively). Meanwhile, the MA/EDTAFeNa-HT2 exhibited a better durability to methanol crossover than the Pt/C catalyst.

## 2. Results and Discussion

We prepared the series of MA/EDTAFeNa-HT2 by the pyrolysis/carbonization (HT1) of MA and EDTAFeNa together with mild acid-leaching (AL), followed by secondary heat-treatment/annealing (HT2) at varied temperature under $N_2$ for 1 h (Scheme 1). The details for the synthetic process are described in the experimental section.

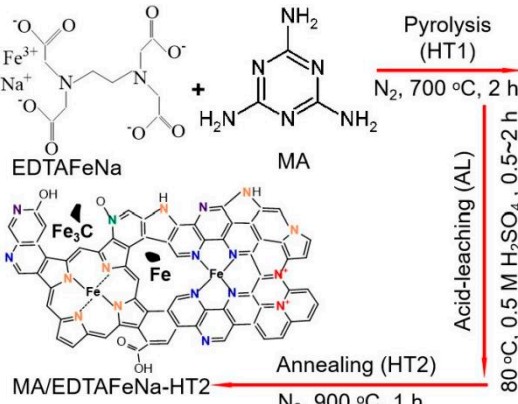

**Scheme 1.** Schematic process of the MA/EDTAFeNa-HT2 catalysts.

Figure 1 shows TEM images of MA/EDTAFeNa-HT2 obtained under different acid-leching times and pyrolysis temperatures (HT2). For the control sample (MA/EDTA-

HT2) without Fe, no particles were observed in the image (Figure 1a). After EDTA was replaced with EDTAFeNa, including Fe, large quantities of nanoparticles were generated (Figure 1b), which should be associated with crystalline iron species. By changing the AL time (Figure 1b–d), the dispersion of iron species appeared to be improved and their particle size became smaller. To increase the annealing temperature (HT2), it seemed to have no obvious effect on the dispersion and size of nanoparticles (Figure 1e–h), which could be rationalized by the fact that the residual Fe species were probably encapsulated into graphitic carbon layers after the first pyrolysis/carbonization and AL, thus avoiding further aggregation during HT2.

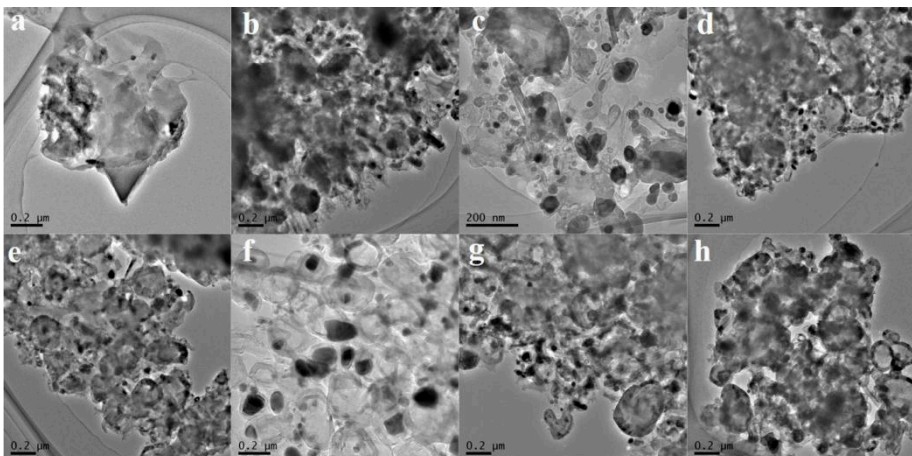

**Figure 1.** TEM images of the MA/EDTA-HT2 (**a**) and MA/EDTAFeNa-HT2 prepared under AL for 0.5 h (**b**), 1 h (**c**) and 2 h (**d**), HT2 at 600 °C (**e**), 700 °C (**f**), 800 °C (**g**) and 1000 °C (**h**), respectively, with other conditions unchanged.

On the powder XRD diffraction patterns shown in Figure 2A, the characteristic peaks were found around at ($2\theta$ = 30.18°, 35.60°, 57.90°, 63.00°), ($2\theta$ = 37.80°, 42.89°, 43.82°, 49.10°, 50.80°, 54.52°) and ($2\theta$ = 44.80°, 65.06°, 82.51°) for all samples, which confirmed the presence of iron species such as $Fe_3O_4$, $Fe_3C$, and metallic Fe, according to the previous reports [29,30]. Even though the AL time was extended until 2 h, there were still strong signals attributed to these iron entities (Figure 2A, b–d). This result revealed that these residual crystalline iron species were encapsulated into carbon layers, thus they were difficult to be removed. The annealing temperature (HT2) (Figure 2A, e–h) and mass ratio of MA to EDTAFeNa (Figure S1) during the investigation range seemed not to influence the type of iron species generated due to it being protected by carbon layers.

The vibrating sample magnetometer (VSM) was employed to measure magnetic characteristics of the series of MA/EDTAFeNa-HT2 materials. As depicted in Figure 2B, the hysteresis loops revealed the presence of magnetic species, produced through pyrolyzing the EDTAFeNa under high temperature and $N_2$ atmosphere, in all the samples except for the MA/EDTA-HT2. As reported elsewhere [31], the saturated magnetization value ($\sigma_s$) is indicative of the content of magnetic species. It can be observed that the $\sigma_s$ decreased with AL time at a certain range; therefore, extending the AL time could remove more external magnetic Fe species, which have been considered to be low active or inactive towards ORR [32,33]. It is noted that the annealing temperature (HT2) almost has no effect on the content of magnetic species after the treatments by the identical HT1 and AL.

Raman spectroscopy (Figure 2C) identified the typical D band (~1350 cm$^{-1}$) and G band (1581 cm$^{-1}$), which are induced by the breathing of carbon atoms close to the edge of a graphene sheet and in-plane motion of the sp$^2$ hybridization carbon atoms in the graphene layer [34], respectively, confirming the existence of defective and graphitic carbon for all the catalysts. Obviously, the series of MA/EDTAFeNa-HT2 samples (b–h) exhibited the lower $I_D/I_G$ (relative intensity ratio of D and G peaks) compared to the MA/EDTA-HT2

(a), which could be attributed to the role of $Fe^{3+}$ in EDTAFeNa as a graphitization catalyst. Extending the AL time favored a slight enhancement in the degree of graphitization, which was probably due to the removal of more Fe species. Moreover, increasing the annealing temperature (HT2) from 600 to 1000 °C had little influence over the carbon structure. Moreover, the intensity of the 2D band has also been considered to reflect the degree of graphitization in carbon materials; the larger the intensity, the higher the level of graphitization [11]. It is apparent that the MA/EDTAFeNa-HT2 samples (c, d, and g) had the most intense 2D bands, while the MA/EDTA-HT2 (a) and MA/EDTAFeNa-HT2 samples (e and f) had the weakest ones, which is in accordance with the $I_D/I_G$ values.

From the XPS survey spectra shown in Figure 2D and Figure S2, the surface elements for the MA/EDTA-HT2 (a) were composed of C, O and N but additional Fe for the series of MA/EDTAFeNa-HT2 materials (b–h), and their atomic percent is determined in Table S1. It can be seen that MA/EDTA-HT2 had a higher N content than the series of MA/EDTAFeNa-HT2 under the identical HT2 (900 °C) and mass ratio of dual precursors, signifying that the presence of $Fe^{3+}$ resulted in more N loss. There is no straightforward correlation between the content of the surface elements and preparation conditions such as AL time and HT2. However, the N content was gradually increased as the mass ratio of MA to EDTAFeNa increased from 1:4 to 4:4, implying that the dual precursors could result in higher N content than the sole EDTAFeNa.

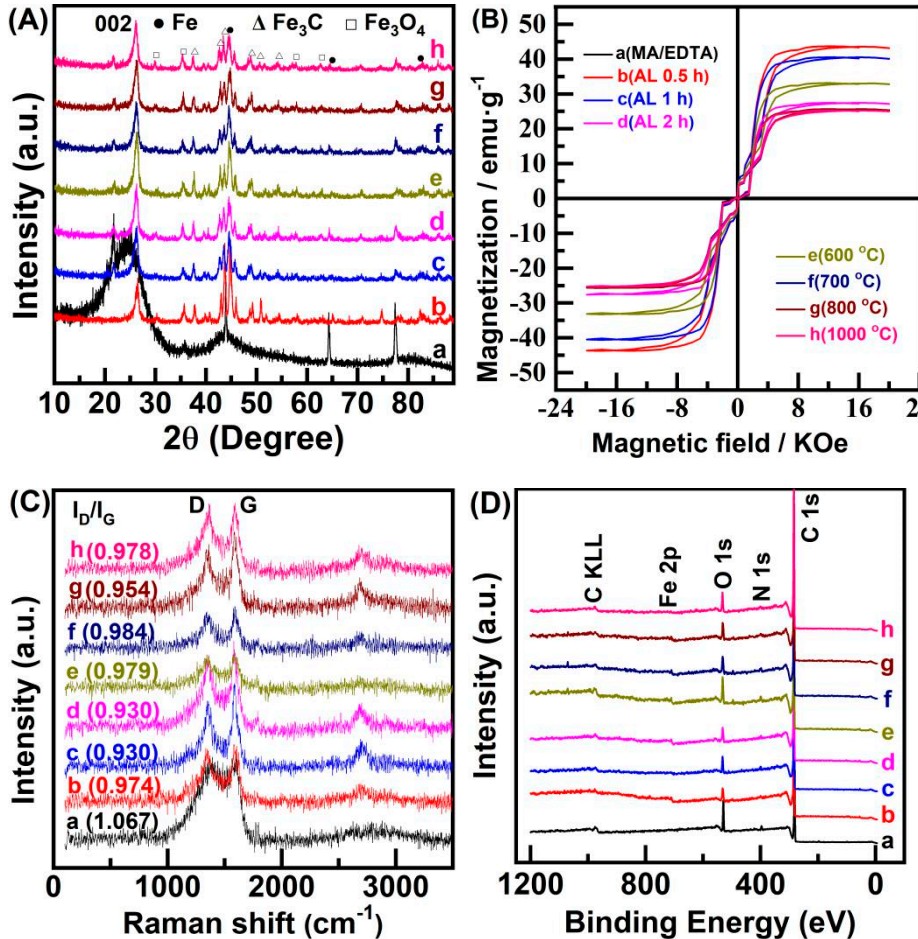

**Figure 2.** (**A**) XRD patterns, (**B**) field-dependent magnetization, (**C**) Raman spectra, and (**D**) XPS survey spectra of the MA/EDTA-HT2 (a) and MA/EDTAFeNa-HT2 prepared under AL for 0.5 h (b), 1 h (c) and 2 h (d), HT2 at 600 °C (e), 700 °C (f), 800 °C (g) and 1000 °C (h), respectively, with other conditions unchanged.

Figure 3 demonstrates the N1s XPS spectra for the series of samples. The deconvoluted peaks centered at binding energies of around 398.4, 399.7, 401.2 and 402.4–404.9 eV can be typically assignable to pyridinic, pyrrolic, graphitic/quaternary and oxidized pyridinic nitrogen atoms, respectively [35]. Note that the N1s signals obviously changed when the EDTA was replaced with EDTAFeNa or the MA was introduced. Specifically, only pyridinic and quaternary N species were found on the spectra of the MA/EDTA-HT2 (a), but four N species for all the MA and EDTAFeNa co-derived samples (b–f), which indicates that the presence of $Fe^{3+}$ promoted the diversification of N species during the carbonization. In addition, there was a marked change in the relative proportions of each species with the increase in mass ratio of MA to EDTAFeNa from 1:4 to 4:4 (c–f). The atomic percentage of each N species was calculated according to their peak area and is listed in Table S2. It is observed that the percent of pyridinic and quaternary N species increased as the mass ratio of MA to EDTAFeNa increased, indicating that the N configurations could be regulated by optimizing dual precursors to a certain degree.

The low-temperature nitrogen adsorption/desorption analyses were utilized to gain an in-depth understanding of the textural properties of the MA/EDTAFeNa-HT2 materials. As plotted in Figure 4A, the $N_2$ volume adsorbed was nearly zero for the MA/EDTA-HT2. The isotherms for the MA and EDTAFeNa co-derived samples, by contrast, took on combined type I/IV curves and distinct H3 hysteresis loops were observed from middle to high relative pressure ($0.46 < P/P_0 < 1.0$). The feature disclosed that both the micro- and mesopores contributed to the porosity in all these samples [36]. The pore size distributions further revealed co-existence of the micro/mesopores and appeared to be narrowly distributed for the mesopores (2~4.5 nm). The sodium in EDTAFeNa could act as an activation agent, resulting in micropores in carbonaceous materials through the evaporation and etching of sodium during pyrolysis [37]. It can also be observed that the annealing temperature (HT2) barely affected the types of isotherms and hysteresis loops (Figure 4B).

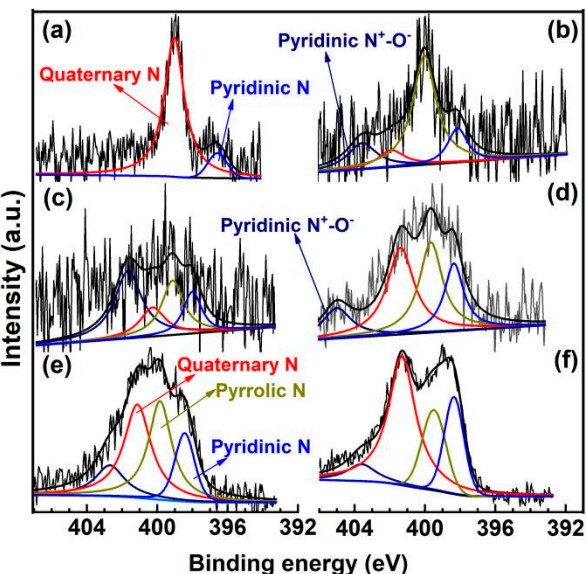

**Figure 3.** XPS N 1s spectra of the MA/EDTA-HT2 (a), MA/EDTAFeNa-HT2 (b and c for AL 0.5 and 2 h, respectively) and MA/EDTAFeNa-HT2 (d, e, and f for mass ratios of MA/EDTAFeNa of 2:4, 3:4, and 4:4, respectively).

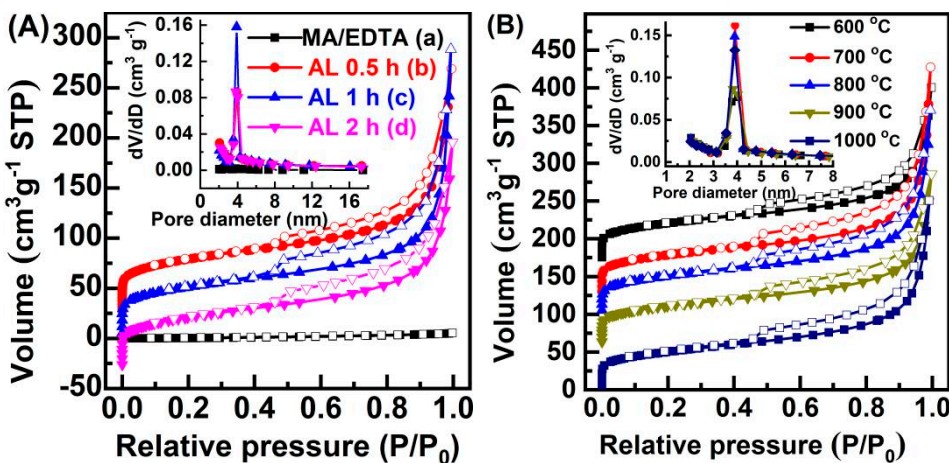

**Figure 4.** N₂ adsorption/desorption isotherms and pore size distribution (inset) for (**A**) the MA/EDTA-HT2 (a), MA/EDTAFeNa-HT2 (b, c, and d for AL 0.5, 1, and 2 h, respectively), and (**B**) MA/EDTAFeNa-HT2 (HT2 for 600, 700, 800, 900 and 1000 °C, respectively). The values of volume adsorbed for the micropore range are not real due to vertical translation of these isotherms for clearer observations.

The structural parameters are determined in Table 1 according to BET and BJH analysis, indicating nearly the same average pore sizes for all samples but a very low pore volume and specific surface area only for the control sample (MA/EDTA-HT2). Moreover, with regard to the previously reported sample (EDTAFeNa-HT2 with 3.9 nm of pore diameter and 295 m² g⁻¹ of specific surface area) [38], the generation of most mesopores was attributed to the removal of Fe species by AL. As a result, it can be concluded that the mesopore size should match with particle size of the generated Fe species. By comparison, the introduction of MA to EDTAFeNa resulted in decrease in the specific surface area and pore volume to a certain degree without influencing the pore diameter. Therefore, it can be concluded that the exposed positions after the removal of some iron species by AL was dominantly responsible for the generation of mesopores and the introduction of MA seemed not to impact on particle sizes of the Fe species generated from EDTAFeNa. The AL time and the annealing temperature (HT2) had little effect on the average pore size and no significant effects on specific surface area of the MA/EDTAFeNa-HT2, which shows that the generated Fe species were easily removed in a short time (0.5 h), and it was considerably stable after the dual precursors were pyrolyzed and acid-leached.

**Table 1.** Textural properties of the MA/EDTAFeNa-HT2 [1].

| Sample | $D_P$ (nm) | $V$ (cm³ g⁻¹) | $S_{BET}$ (m² g⁻¹) |
|---|---|---|---|
| MA/EDTA-HT2 | 3.8 | 0.01 | 4 |
| MA/EDTAFeNa-HT2 (A 0.5 h) | 3.9 | 0.36 | 196 |
| MA/EDTAFeNa-HT2 (AL 1 h) | 3.9 | 0.44 | 239 |
| MA/EDTAFeNa-HT2 (AL 2 h) | 3.8 | 0.33 | 180 |
| MA/EDTAFeNa-HT2 (600 °C) | 4.0 | 0.34 | 182 |
| MA/EDTAFeNa-HT2 (700 °C) | 3.9 | 0.46 | 255 |
| MA/EDTAFeNa-HT2 (800 °C) | 3.9 | 0.42 | 227 |
| MA/EDTAFeNa-HT2 (1000 °C) | 3.9 | 0.39 | 207 |

[1] $D_P$: average pore diameter obtained by desorption isotherm; $V$: pore volume; $S_{BET}$: BET specific surface area.

Electrocatalysis for ORR on dual precursor-derived MA/EDTAFeNa-HT2 was investigated by using RDE via LSV measurements in Figure 5A. To compare their ORR activity more intuitively, from these LSV curves the parameters such as $E_{onset}$, $E_{1/2}$ and limiting current density ($J_L$) were extracted into Table 2. Generally, the larger the parameters, the

higher the corresponding ORR activity. Obviously, the MA/EDTA-HT2 (a) exhibited the lowest ORR activity, which was attributed to the absence of some active iron species and ultra-low specific surface area. When EDTA was replaced with EDTAFeNa as a precursor, varied iron moieties including active ones ($FeN_x$) [39,40] were produced and the amount of mesopores and surface area were also greatly enhanced by removing some iron nanoparticles. As a result, their ORR activity was sharply improved (Figure 5A, b–g), confirming that some active iron species and large surface area resulting from mesopores played positive roles in ORR. The AL time was controlled to regulate the remaining content of iron species and surface area for the MA/EDTAFeNa-HT2 (Figure 5A, b–d). It was found that the ORR activity was obviously boosted when the AL time was extended from 0.5 to 1 h (from b to c). However, the ORR activity seemed not to be further improved by prolonging the AL time to 2 h (d), which indicated that 1~2 h of AL was enough and applicable for removing inactive Fe species, obtaining mesopores and large specific surface area, and increasing the accessibility of exposed active sites to oxygen, resulting in high ORR activity.

The effect of the annealing temperature (HT2) on ORR activity was also investigated (Figure S3). The results agreed with the previous reports [16,19,41,42] that the optimum temperature fell in the range of 800 to 1000 °C. The observed differences in ORR activities should be because of the presence and density of active sites and maybe also the morphological properties due to similar mesoscopic structure (pore size, specific surface area), inactive crystalline Fe species and content, and graphitization degree for the series MA/EDTAFeNa-HT2 prepared under varied annealing temperature. To keep AL time (2 h), HT2 (900 °C) and the other experimental conditions unchanged, the mass ratio of MA to EDTAFeNa was studied as well (Figure 5A, e–g). After the introduction of MA to EDTAFeNa as a dual precursor, the $E_{1/2}$ for the as-prepared MA/EDTAFeNa-HT2 (d, e, f and g) had a positive shift of 20~50 mV relative to that for only the EDTAFeNa-derived counterpart (EDTAFeNa-HT2). Overall, the ORR activity of the obtained MA/EDTAFeNa-HT2 presented increasing tendency with the increased mass ratio. Together with characterization analysis, effects of the variety of inactive crystalline iron species and mesoscopic structure on ORR activity could be nearly excluded; thus, the higher pyridinic and quaternary N species are considered to be responsible for the enhancement of ORR activity due to a possible increase in the number of the active $CN_x$ and $FeN_xC_y$ moieties involving N and Fe species. It is also noted that, in terms of $E_{onset}$, $E_{1/2}$ and kinetic current density ($J_k$), the ORR activity on the MA/EDTAFeNa-HT2 (g) was the similar to that on the state-of-the-art Pt/C catalyst.

The catalytic performance is also closely associated with catalyst selectivity, namely, the electron transfer mechanism during the electrocatalysis for ORR. The electron transfer number (*n*) per $O_2$-reduction process was calculated based on the Koutecky−Levich (K–L) plots deriving from LSV curves under different rotation rates (Figure 5B–D) at applied electrode potentials [43]. The insets in Figure 5B reveal that $H_2O_2$ was mainly produced during ORR which occurred on the MA/EDTA-HT2 via a two-electron process [44]. In contrast, the ORR was catalyzed to predominantly produce $H_2O$ through a four-electron pathway by MA/EDTAFeNa-HT2, which is similar to that by the Pt/C (JM). In the proposed two-electron mechanism, $O_2$ molecules adsorb at the N-doped active carbon atoms ($O^*_2$) as the first step of the ORR followed by transfer of proton-electron to the adsorbed $O_2$, forming the *OOH intermediate, which is then bonded with another proton to produce $H_2O_2$ and detach from the active site; however, in the four-electron pathway, the produced *OOH intermediate is followed by the second electron transfer in the form of hydrogen addition again to release a water molecule, giving rise to *O species at active sites involving Fe species, and then two-electron transfer in the form of hydrogen addition again to the formed *O with desorption of a second water molecule [28,45]. In retrospect of characterization analysis, it was inferred that both the functional configuration involving active Fe species and abundant micro/mesoporous structure are responsible for four-electron selectivity towards ORR.

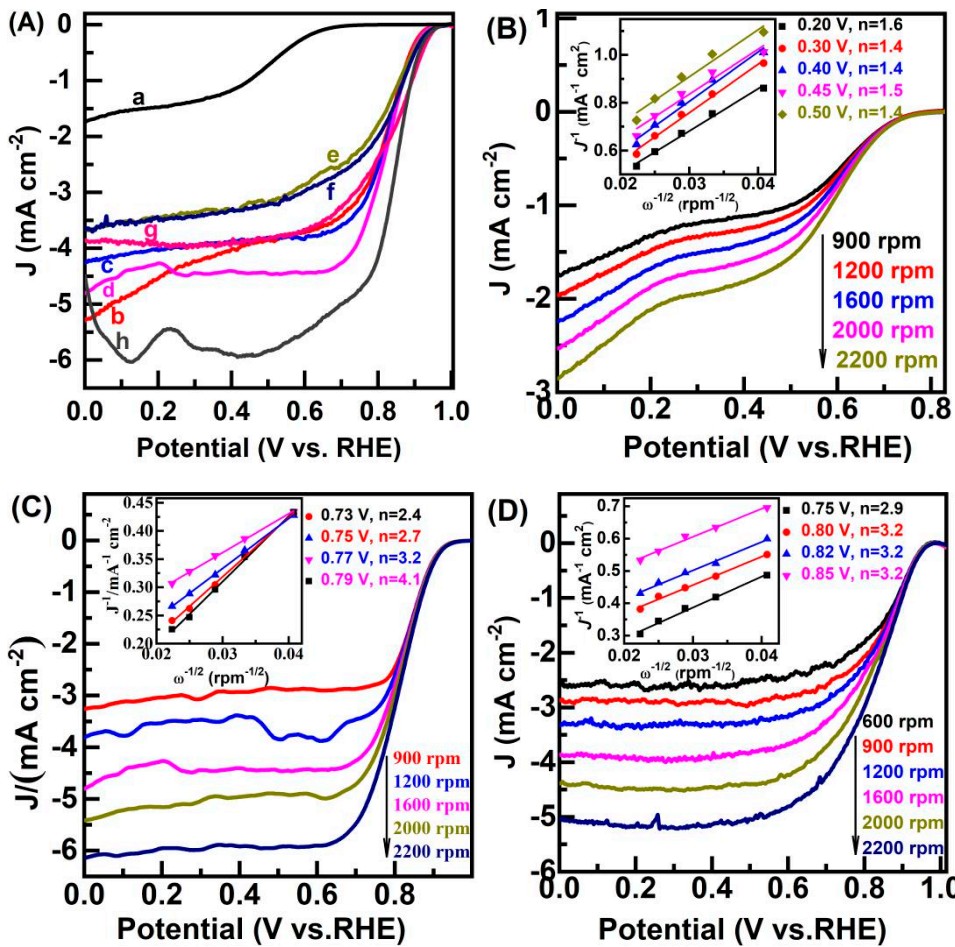

**Figure 5.** (**A**) LSV curves for ORR on the MA/EDTA-HT2 (a), MA/EDTAFeNa-HT2 (b, c and d for AL 0.5, 1 and 2 h, respectively), MA/EDTAFeNa-HT2 (e, f and g for mass ratio of MA/EDTAFeNa of 2:4, 3:4 and 4:4, respectively) and Pt/C (Johnson Matthey, 20 wt.% Pt) (h) in 0.1 M KOH (scanning and rotation rates for 10 mV s$^{-1}$ and 1600 rpm, respectively); polarization curves at various rotation rates of (**B**) the MA/EDTA-HT2 (a), (**C**) MA/EDTAFeNa-HT2 (d, AL 2 h) and (**D**) MA/EDTAFeNa-HT2 (g, 4:4) with the corresponding K–L plots (J$^{-1}$ vs. $\omega^{-1/2}$) at the applied electrode potentials (insets in B–D, respectively).

**Table 2.** Performances of ORR on the MA/EDTAFeNa-HT2 and Pt/C catalyst in 0.1 M KOH [1].

| Simple | $E_{onset}$ (V) | $E_{1/2}$ (V) | $J_L$ (mA cm$^{-2}$) | $J_k$ (mA cm$^{-2}$) | |
|---|---|---|---|---|---|
| | | | | 0.85 V | 0.87 V |
| MA/EDTA-HT2 (a) | 0.71 | 0.49 | 1.51 | - | - |
| MA/EDTAFeNa-HT2 (b) | 0.95 | 0.79 | 3.94 | 1.40 | 0.93 |
| MA/EDTAFeNa-HT2 (c) | 0.96 | 0.83 | 3.81 | 2.20 | 1.23 |
| MA/EDTAFeNa-HT2 (d) | 0.96 | 0.82 | 4.45 | 2.20 | 1.22 |
| MA/EDTAFeNa-HT2 (e) | 0.97 | 0.81 | 3.37 | 1.68 | 1.10 |
| MA/EDTAFeNa-HT2 (f) | 0.98 | 0.83 | 3.43 | 2.21 | 1.40 |
| MA/EDTAFeNa-HT2 (g) | 0.98 | 0.84 | 3.94 | 3.26 | 2.27 |
| Pt/C (JM, 20 wt.% Pt) (h) | 0.97 | 0.84 | 5.92 | 4.48 | 2.31 |
| EDTAFeNa-HT2 | 0.93 | 0.79 | 4.58 | 0.95 | 0.49 |

[1] Potentials were normalized to RHE and $J_k$ values were obtained based on the K–L equation. a–h are the sample numbers, shown in Figure 5.

The Tafel plots of mass transport-corrected currents, deduced from Figure 5A, for the MA/EDTAFeNa-HT2 and Pt/C (JM) catalysts towards ORR in 0.1 M KOH are demon-

strated in Figure 6A. They had Tafel slopes (TS) of 46.8 mV/decade and 56.3 mV/decade within the identical potential region (0.88 < E < 0.95 V) [1], respectively, which indicates slightly smaller barriers to electron transport during ORR on the MA/EDTAFeNa-HT2 than on the Pt/C (JM) catalyst. However, a rapid shift of the TS from ~60 to ~120 mV/decade at the selected potential range (0.82 < E < 0.95 V) was observed for both of them, reflecting possibly similar ORR kinetics that the initial electron transfer in addition of hydrogen to the adsorbed $O_2$ molecules was the rate-limiting step [46]. The TS values revealed that the MA/EDTAFeNa-AL-HT2 had comparable ORR activity to that of the Pt/C catalyst.

The durability against methanol crossover of the MA/EDTAFeNa-HT2 and Pt/C was investigated during electrocatalytic ORR. As depicted in Figure 6B, the ORR on the Pt/C was immediately seriously disturbed once the methanol was introduced, which is attributed to the fact that methanol is easily oxidized by Pt. As a result, the ORR current gradually decreased to ~60% of its initial value. In contrast, the ORR on the MA/EDTAFeNa-HT2 was not significantly affected by the addition of methanol and 79% of its initial current was still retained. This result showed that the present MA/EDTAFeNa-HT2 exhibited more outstanding methanol tolerance than the Pt/C.

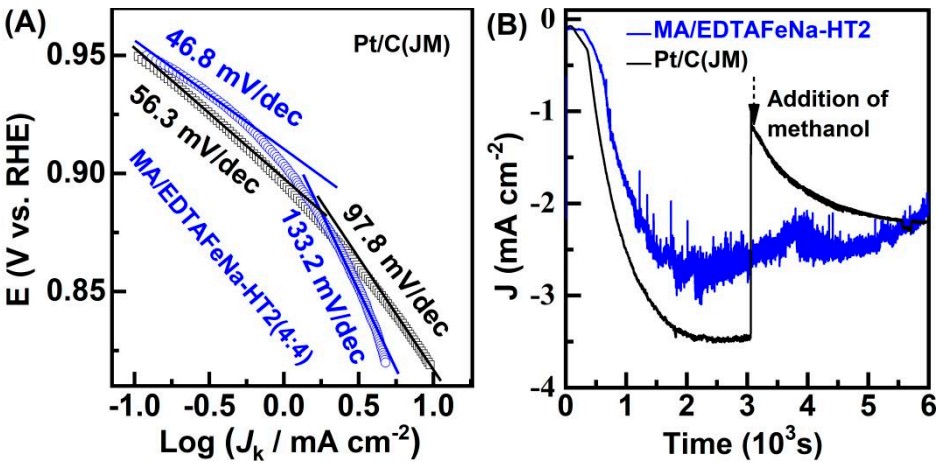

**Figure 6.** (**A**) Tafel plots and (**B**) resistance to methanol crossover for MA/EDTAFeNa-HT2 (4:4) and Pt/C catalyst (JM, 20 wt.% Pt).

## 3. Materials and Methods

### 3.1. Materials

EDTA ferric sodium salt (EDTAFeNa, 99.0%) was purchased from Tianjin Bodichem Co. Ltd. (Tianjin, China). Melamine (99.0%) was obtained from Tianjin Beilian Fine Chemicals Development Co. Ltd. (Tianjin, China). Pt/C catalyst (20 wt.% Pt) was purchased from Johnson Matthey (Shanghai) Chemical Co., Ltd. (London, UK). Nafion (5% wt.%) was obtained from Du Pont Co. Ltd. (Wilmington, DE, USA). All reagents were used as received without further purification.

### 3.2. Catalyst Preparation

EDTAFeNa and MA co-derived MA/EDTAFeNa-HT2 catalysts were synthesized by a modified method reported in the previous literature [13]. Briefly, 2 g EDTAFeNa and 0.5 g melamine (MA) were mixed, carbonized (pyrolyzed) at 700 °C for 2 h with a ramp rate of 10 °C min$^{-1}$ from room temperature under flowing nitrogen, and then naturally cooled to room temperature. Subsequently, the resultant black powder was acid-leached for 0.5 h at 80 °C in 30 mL 0.5 M $H_2SO_4$ solution under stirring, and then washed with deionized water and dried in an oven overnight. Finally, the sample was heat-treated again at 900 °C for 1 h with a heating rate of 5 °C min$^{-1}$ under flowing nitrogen and then naturally cooled to room temperature. At the final stage, the as-obtained sample was denominated as MA/EDTAFeNa-HT2.

Firstly, the acid-leaching time (1 and 2 h) was investigated under the same preparation conditions as described above. Secondly, the final heat-treated temperature (600, 700, 800 and 1000 °C) was investigated under identical experimental conditions except for the acid-leaching time which was changed to 2 h. Thirdly, to maintain the acid-leaching time and final annealing temperature as 2 h and 900 °C, respectively, with other conditions unchanged, the control samples without iron species were also prepared identically, except for EDTAFeNa which was replaced with EDTA as a precursor (denoted as MA/EDTA-HT2). Lastly, the mass ratio of MA to EDTAFeNa (2:4~4:4) was also studied under the same conditions as the control sample.

### 3.3. Characterization of Catalysts

The transmission electron microscopy (TEM) (FEI, Tecnai G2 F30 S-TWIN, Portland, OR, USA) was used to observe morphologies of these samples. X-ray diffraction (XRD, XRD-6000, Shimadzu Co., Kyoto, Japan) with Cu K$\alpha$ radiation was used to distinguish configurations of iron species. An automatic physisorption apparatus (3H-2000PM1, Bechtel instrument technology (Beijing) Co. Ltd., Beijing, China) was used to determine the surface area, pore size, and pore volume based on $N_2$ adsorption–desorption isotherms, BET and BJH analysis. X-ray photoelectron spectroscopy (XPS, ESCALAB220-IXL, Thermo Fisher Scientific, Waltham, MA, USA) was utilized to measure surface elementary composition. Raman spectroscopy (DXR Raman Microscope, JY-HR800 micro-Raman, Thermo Fisher Scientific, Waltham, MA, USA) with a 532 nm wavelength YAG laser was used to probe carbon structure. A Vibrating Sample Magnetometer (VSM, EV9, MicroSense, Milpitas, CA, USA) was used to test saturation magnetization.

### 3.4. Electrochemical Measurements

The ink was prepared by adding 3 mg catalyst into 0.5 mL of 0.5 wt.% nafion solution (6 μg μL$^{-1}$) under ultrasonic treatment for 0.5 h. Afterwards, 15 μL of the as-prepared catalyst ink was dropped onto the surface of a well-polished glassy carbon electrode (GCE) and placed under infrared light until dry. The resulted catalyst-loaded (459 μg cm$^{-2}$) GCE was measured as a working electrode for ORR in a three-electrode cell; the other two electrodes (reference and counter electrodes) were Ag/AgCl (in 3.5 M KCl) and a Pt wire, respectively.

Cyclic voltammetry (CV) measurements were performed in $O_2$-saturated 0.1 M KOH from −0.2~1.0 V with a scan rate of 50 mV s$^{-1}$ by using a rotating disk electrode (RDE) as the working electrode for 30 cycles, followed by linear sweep voltammetry (LSV) measurements from approximately −0.2 to 1.0 V with a scan rate of 10 mV s$^{-1}$ at varied rotating speeds from 600 to 2200 rpm. The measurements for CV and LSV in $N_2$-saturated 0.1 M KOH were also performed as the control experiments under the identical conditions with those in $O_2$.

The chronoamperometric assessments for tolerance against methanol crossover of the MA/EDTAFeNa-HT2 and Pt/C catalysts were carried out at 0.85 V in $O_2$-saturated 0.1 M KOH. The RDE rotation speed and the concentration of methanol added were set for 1600 rpm and 3 M in the electrolyte, respectively.

## 4. Conclusions

In summary, we developed a facile and economical route to successfully synthesize self-supported Fe-N-C materials (denoted as MA/EDTAFeNa-HT2) where EDTA ferric sodium (EDTAFeNa) and melamine (MA) were utilized as the dual precursors. After optimization of the preparation processes, the as-prepared MA/EDTAFeNa-HT2 catalyst exhibited comparable ORR activity at the low overpotential region and a similar 4e$^-$ dominant transfer pathway but superior methanol tolerance to 20 wt.% Pt/C (JM) in alkaline media. In the range investigated, the acid-leaching time and final annealing temperature (HT2) seemed to have little influence on varieties of the generated crystalline iron species, mesostructure or degree of defectiveness or graphitization, but some influence on the

amount of residual magnetic Fe species of the obtained MA/EDTAFeNa-HT2. Furthermore, the introduction of MA had almost no effect on varieties of the generated crystalline iron species but an obvious effect on the total nitrogen content, which increased with increases in the mass ratio of MA to EDTAFeNa. Moreover, the percentages of the pyridinic and quanternary nitrogen also increased accordingly. It was found that content of the magnetic Fe species and the pyridinic/quanternary nitrogen was negatively and positively associated with the ORR activity of the MA/EDTAFeNa-HT2, respectively. The correlation indicates that less crystalline/magnetic iron species and increased pyridinic/quanternary nitrogen content could facilitate formation of a higher number of active sites. A dual precursor strategy based on EDTAFeNa opens up a potential avenue to concurrently engineer porous structure, varieties of Fe species, nitrogen configurations, and the content of self-supported Fe-N-C catalysts for further enhancement of ORR performance.

**Supplementary Materials:** The following are available online at https://www.mdpi.com/article/10.3390/catal11050623/s1, Figure S1: XRD patterns of the MA/EDTA-HT2 (a) and MA/EDTAFeNa-HT2 prepared under 0.5 h (b), 1 h (c) and 2 h (d) of acid-leaching time, and mass ratio of MA/EDTAFeNa of 2:4 (e), 3:4 (f) and 4:4 (g), respectively; Figure S2: XPS survey spectra of the MA/EDTA-HT2 (a) and MA/EDTAFeNa-HT2 prepared under 0.5 h (b), 1 h (c) and 2 h (d) of acid-leaching time, and mass ratio of MA/EDTAFeNa of 2:4 (e), 3:4 (f) and 4:4 (g), respectively; Figure S3: LSV curves for ORR on the MA/EDTAFeNa-HT2 prepared under different pyrolysis temperature (HT2), respectively; Table S1: Surface chemical compositions and atomic concentration of the series of MA/EDTAFeNa-HT2; Table S2: Atomic concentration of varied N configurations in the series of MA/EDTAFeNa-HT2.

**Author Contributions:** Conceptualization, M.S. and X.-H.Y.; methodology, X.-H.Y.; validation, M.S., Z.M., T.X., H.S. and X.-H.Y.; formal analysis, M.S., Z.M., T.X., H.S. and X.-H.Y.; investigation, M.S. and Z.M.; resources, M.S., Z.M., T.X., H.S. and X.-H.Y.; writing—original draft preparation, M.S., Z.M. and X.-H.Y.; writing—review and editing, T.X., H.S. and X.-H.Y.; supervision, X.-H.Y.; project administration, X.-H.Y.; funding acquisition, X.-H.Y. All authors have read and agreed to the published version of the manuscript.

**Funding:** This research was funded by West Light Foundation of The Chinese Academy of Sciences (No. XAB2019AW10), the Natural Science Foundation of Ningxia Province (No. 2019AAC03114), the National Natural Science Foundation of China (No. 21563001) and the Graduate Student Innovation Project of North Minzu University (No. YCX20127).

**Acknowledgments:** The authors would also like thank Hu Zhun at Xi'an Jiaotong University for help with sample characterizations.

**Conflicts of Interest:** The authors declare no conflict of interest. The funders had no role in the design of the study; in the collection, analyses, or interpretation of data; in the writing of the manuscript, or in the decision to publish the results.

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
