# Peer review of "Electrocatalysis for Oxygen Reduction Reaction on EDTAFeNa and Melamine co-Derived Self-Supported Fe-N-C Materials"

_catalysts, doi:10.3390/catal11050623_

Round 1

Reviewer 1 Report

The authors report on PGM-free Fe-N-C electrocatalysts for ORR. Their approach for Fe-N-C catalysts synthesis is not very novel, however the obtained results are interesting and hold some scientific value. Generally, the topic of PGM-free electrocatalysts for PEFC is very interesting and Fe-N-C (and Co-N-C) structures are the most promising among all the studied PGM-free catalysts. The manuscript is interesting and authors utilized many advanced techniques to characterize the obtained materials, I especially appreciate that they used the vibrating sample magnetometer (VSM) to assess magnetic response of the samples. Here are some hints how to improve the manuscript:

  1. English must be improved, please correct carefully 
  2. It seems that the authors differentiate between the term carbonization and pyrolysis (as they did in scheme 1). Carbonization is just a specific example of pyrolysis, but these are the same phenomena. The second heating could be named annealing (for instance).
  3. Why was the catalytic performance studied only in alkaline environment and not in acidic conditions? Please try to justify this.
  4. As an example of a template-free strategy you can also mention Carbon 116 (2017) 655-669, 10.1016/j.carbon.2017.02.045
  5. I would encourage the authors to include the 2D bands in the discussion concerning Raman spectra. Intense 2D bands for sample c, d, g indicate high level of graphitization, in accordance with the ID/IG values.
  6. About Figure 4, there might be an error in presentation. For instance, according to Fig. 4B sample MA/EDTAFeNa-HT2 600’C has the highest volume adsorbed [cm3/g] up to 200 for the micropore range. But its S-BET is only 182 m2/g. Please correct the figure or give additional appropriate explanation in the text or under figure caption.
  7. Table 1. – there is no need to give the values of S-BET with the precession up to 0.1 m2/g. These evaluations are never that accurate, they are more like +/- 10%. For instance, instead of 239.3 it is better to give 239 m2/g or even 240 m2/g.
  8. Figure 5. please in the figure caption add the solution characteristic (0.1 M KOH?). The same comment for Table 2.
  9. What could be the role of sodium during pyrolysis? As known, sodium can act as activation agent increasing porosity of carbonaceous materials?
  10. After reading the conclusions one can get impression that most of the reported results were easy to predict or were expected. In the conclusion the novelty and new important discoveries should be stresses.

Author Response

General response: We appreciate your positive comments and good advice, which have benefited us greatly in the field of Fe-N-C materials and really have helped us in improving the quality of our manuscript. Followed are our responses to your specific comments.

  1. 1.English must be improved, please correct carefully.

Authors’ response: Thanks. We have done our best to correct carefully.

  1. It seems that the authors differentiate between the term carbonization and pyrolysis (as they did in scheme 1). Carbonization is just a specific example of pyrolysis, but these are the same phenomena. The second heating could be named annealing (for instance).

Authors’ response: Thanks for sound advice. The first and second heat-treatments have been uniformly named pyrolysis and annealing, respectively.

  1. Why was the catalytic performance studied only in alkaline environment and not in acidic conditions? Please try to justify this.

Authors’ response: This is very insightful. It is reported that various Fe-N-C materials have exhibited remarkable ORR activity in alkaline electrolyte and even outperformed the Pt/C catalyst; however, they showed moderate to low ORR activity in more challenging acidic electrolyte. The comparisons indicate that oxygen molecules are more easily activated and the Fe-N-C materials are more stable in alkaline electrolyte than in acidic one. In this case, I think the ORR carried out in alkaline electrolyte is always first employed to evaluate electrocatalytic activity of the Fe-N-C materials. After all, it is relatively easy to make the Fe-N-C materials rival and even outperform the Pt/C catalyst, which is always encouraging. In fact, we have evaluated ORR activity of the MA/EDTAFeNa-HT2(4:4) in acidic electrolyte, which was much lower than that in alkaline electrolyte. In our following work, preparations of the Fe-N-C materials derived from EDTA and hard templates are further optimized for enhancing ORR activity in acidic electrolyte.  

  1. As an example of a template-free strategy you can also mention Carbon 116 (2017) 655-669, 10.1016/j.carbon.2017.02.045

Authors’ response: Thanks for your recommendation. The literature has benefited us greatly in this field of a template-free strategy and materials characterization. We have added them as reference 11.

  1. I would encourage the authors to include the 2D bands in the discussion concerning Raman spectra. Intense 2D bands for sample c, d, g indicate high level of graphitization, in accordance with the ID/IG values.

Authors’ response: Thanks for the good suggestion. The discussion about 2D bands has been supplemented to the revised manuscript (Line 128-134). In fact, we learned the knowledge about 2D bands from reference 11.

  1. About Figure 4, there might be an error in presentation. For instance, according to Fig. 4B sample MA/EDTAFeNa-HT2 600’C has the highest volume adsorbed [cm3/g] up to 200 for the micropore range. But its S-BET is only 182 m2/g. Please correct the figure or give additional appropriate explanation in the text or under figure caption.

Authors’ response: Thank you for friendly reminding. We are sorry for not giving explanation under figure caption. In fact, these isotherms have been vertically translated for the clearer observations. Thus, the MA/EDTAFeNa-HT2(600oC) has not the highest volume adsorbed. The data in Table 1 were checked again, which are right. The explanation has been given under the caption of Figure 4 (Line 186-188)

  1. Table 1. – there is no need to give the values of S-BET with the precession up to 0.1 m2/g. These evaluations are never that accurate, they are more like +/- 10%. For instance, instead of 239.3 it is better to give 239 m2/g or even 240 m2/g.

Authors’ response: Thanks. We agree with your viewpoints. These values have been corrected in Table 1.

  1. Figure 5. please in the figure caption add the solution characteristic (0.1 M KOH?). The same comment for Table 2.

Authors’ response: Thank you. The ORR was catalyzed in 0.1 M KOH, which has been added in the Figure 5 caption and table 2 header.

  1. What could be the role of sodium during pyrolysis? As known, sodium can act as activation agent increasing porosity of carbonaceous materials?

Authors’ response: Thanks for friendly reminding. I think you are right. It is also reported that the smaller pores in carbonaceous materials should come from the evaporation and etching of sodium (Chin. J. Chem., 2018, 36: 287). The role of sodium during pyrolysis has been supplemented in revised manuscript (Line 173-176).

  1. After reading the conclusions one can get impression that most of the reported results were easy to predict or were expected. In the conclusion the novelty and new important discoveries should be stresses.

Authors’ response: Thanks for good advice. We tried to stress the novelty and new important discoveries ((Line 369-374).

All changes were highlighted in yellow background.

Reviewer 2 Report

In this work, Shen et al. studied the influence of melamine and EDTA as dual precursors for the synthesis of Fe-N-C electrocatalysts. They applied a series of different characterization techniques and investigated the activity and selectivity towards the oxygen reduction reaction in alkaline media. While the manuscript is well structured, the interpretation of the data needs intense revision.

Please find my general and specific comments below.

  • Purpose of the introduction of the dual precursor principle is an increase in nitrogen content in the catalysts. While the reader has no comparison for a Fe-N-C made from a single N-precursor, the aim of this model should be a higher number of active sites. One cannot directly correlate a higher N-content to a higher density of active FeNx sites.
  • Generally, the AL conditions might be to mild? In literature, higher concentrated acid is used often times (for example 2M H2SO4).
  • The authors used a Pt wire as CE in their measurements. How about the influence of Pt contaminations to the WE? Additional measurements using a graphite CE are necessary.
  • Due to the relatively high iron content: How do the CVs in N2 atmosphere look like? Are there specific peaks?
  • Generally, many LSVs show strongly varying current densities in the limiting current density region (even the Pt/C reference does not show a “straight line”). Why is that?
  • Values for Eonset and E1/2 are shown and discussed. How exactly were they determined? To me these values do not seem to be very trustworthy as the LSVs show strongly varying trends. I would suggest to rather determine mass activities instead.
  • All samples show iron particles (maybe in slightly different distributions or particles sizes), however, all of them are known to be inactive and undesired for the ORR. So, the observed differences in ORR activities must be due to the presence and density of active sites and maybe also the morphological properties. To me, this point is not discussed in the manuscript.

Line 19/20: only EDTAFeNa-derived counterpart? Where to find this in the presented results?

Line 82: scheme 1: What is the duration of AL?

Line 99: I would not call a standard lab-based XRD pattern wide angle XRD. To me wide angle diffraction can only be measured at the synchrotron.

Line 100/101: The used units look weird, please use: ° 2q (holds for the whole manuscript + SI)

Line 112: how is the saturated magnetization value determined? Where to find it in the graph?

Line 149: MA/EDTATeNa ??

Line 168: Fig 3: I would show a y-axis as it might “calm” the figure.

Line 172: Fig 4: the denotation of the graphs is unfortunate. There is no a, b, c etc found in the figures.

Line 159: What does „no N2 uptake“ mean?

Line 181-184: How can you get from BET surface area to Fe particle size? The conclusion is far from clear to me.

Line 186: How can one know that the generated Fe species were easily to be removed in this argumentation?

Line 199: How do you know you have FeNx sites? There is no direct proof.

Line 216: You cannot exclude the influence of Fe-Nx sites, because you have no proof or quantification for these sites.

Line: I would say that Pt outperforms the Fe-N-C catalysts from this study, especially if you would determine the mass activities of each catalyst.

Line 223/224: Two consecutive one-electron steps? This is wrong.

Line 226-228: This sentence is very general and has no distinct meaning or sense with respect to the presented data. Please specify and clarify.

Line 252: What potential was applied for the MeOH crossover measurements?

Line 255-257: What about the remaining currents after methanol crossover? How were these valued determined?

Line 306/307: Loading in wrong unit? Rather µg/cm2

Line 321: What is economical about this route? This point was not discussed in the manuscript at all.

Author Response

General response: We appreciate your positive comments and good advice, which have benefited us greatly in the field of Fe-N-C materials and really have helped us in improving the quality of our manuscript. Followed are our responses to your specific comments.

  1. Purpose of the introduction of the dual precursor principle is an increase in nitrogen content in the catalysts. While the reader has no comparison for a Fe-N-C made from a single N-precursor, the aim of this model should be a higher number of active sites. One cannot directly correlate a higher N-content to a higher density of active FeNx

Authors’ response: This is very insightful. As you’ve said, the aim of this model is to increase number of active sites. I am sorry for not clearly describing the aim. Although active sites in Fe-N-C materials are not clear, the CNx and FeNxCy moieties involving N species have been proposed as active sites. Based on this, increasing N content is a potential strategy to increase the numbers of active sites. Only EDTAFeNa was employed as precursor, the N content was as low as 0.85 at%; when EDTAFeNa and Melamine (MA) were employed as dual precursors, not only the N content reached as high as 4.41 at% and also the dispersion of Fe species was improved, which could increase density of active CNx and FeNx sites. As a result, the ORR activity was enhanced. We have modified a part of statement in “Introduction” (Line 70-74).

  1. Generally, the AL conditions might be to mild? In literature, higher concentrated acid is used often times (for example 2M H2SO4).

Authors’ response: Thanks for advice. Compared to 2 M H2SO4, the AL conditions (0.5 M H2SO4) is really too mild. The 0.5 M H2SO4 was utilized to acid-leach non-active Fe species simply because Zelenay et al. (Science, 2011, 332, 443) had used this concentration in acid-leaching non-active Fe species of their Fe-N-C catalysts. We will apply H2SO4 with different concentrations to AL treatments in our following work.

  1. The authors used a Pt wire as CE in their measurements. How about the influence of Pt contaminations to the WE? Additional measurements using a graphite CE are necessary.

Authors’ response: Thanks for the good suggestion. Previously, we compared the Pt wire and graphite as CE, respectively, under the identical measurements and found there was little difference in ORR activity. We have just repeated experiments using Pt wire and graphite as CE, respectively, to confirm if there is influence of Pt contaminations on the WE. The results are as follows, which indicate that there are still little difference between Pt wire and graphite CEs whether MA/EDTAFeNa-HT2(4:4) or unpublished Fe-N-C catalyst was used as WE.

  1. Due to the relatively high iron content: How do the CVs in N2atmosphere look like? Are there specific peaks?

Authors’ response: Thank you. The CV curves in 0.1 M KOH look like quasi-rectangle shape. Really, there is no specific peaks on the CVs (Left). However, it should be pointed out that a pair of well-developed redox peaks emerged at ca. 0.64 V on the CVs in a N2-saturated 0.5 M H2SO4 for Fe-N-C catalysts. The redox peaks are believed to be a reversible one-electron process involving Fe3+/Fe2+ couple (Right) (Chin. J. Catal., 2013, 34, 1992).

  1. Generally, many LSVs show strongly varying current densities in the limiting current density region (even the Pt/C reference does not show a “straight line”). Why is that?

Authors’ response: Thanks. This is very insightful. The strong varying current densities in the limiting current density region are mainly attributed to the Rotating Disk Electrode (RDE) due to instability of rotating bar. I am sorry for that our research funds could only afford a cheap RDE (ATA-1B) from Jiangsu Jiangfen Electroanalytical Instrument Co., Ltd. at that time. Until last month of this year, we purchased a RDE710 from Gamry Instruments at last. We found that the fluctuation in current density disappeared entirely when using RDE710.  

  1. Values for Eonsetand E1/2 are shown and discussed. How exactly were they determined? To me these values do not seem to be very trustworthy as the LSVs show strongly varying trends. I would suggest to rather determine mass activities instead.

Authors’ response: Thank you for friendly reminding. The Eonset is defined as the potential at which the ORR generates a current density of −0.02 mA/cm2 (J Power Sources, 2010, 195: 5947) and the E1/2 is defined as the potential at which the current density is half of the limiting current density (JL), which is obtained in the plateau region (red circle in the Figure A). The non-precious metal catalysts (NPMCs) are much cheaper than Pt/C catalyst, which could be at least a main cause why the mass activity data for NPMCs have usually been not reported in the literature. In addition, mass specific activity is defined as normalization of kinetic current (ik) to catalyst mass, in which the limiting current (id) is still need to be used.

  1. Allsamples show iron particles (maybe in slightly different distributions or particles sizes), however, all of them are known to be inactive and undesired for the ORR. So, the observed differences in ORR activities must be due to the presence and density of active sites and maybe also the morphological properties. To me, this point is not discussed in the manuscript.

Authors’ response: Thanks. We strongly agree with you about the analysis in activity difference. As we know, some iron particles are always inevitable during pyrolysis of Fe-containing precursor for Fe-N-C materials; however, unfortunately, it is quite difficult to remove these crystalline iron particles due to being encapsulated by graphitic carbon layers. We also think that differences in ORR activities should be due to the presence and density of active sites and maybe also the morphological properties. This point has been discussed in the revised manuscript (Line 224-231).

Line 19/20: only EDTAFeNa-derived counterpart? Where to find this in the presented results?

Authors’ response: Thank you for friendly reminding. We are sorry for only mentioning it in N2 adsorption/desorption analysis (line 191-195). the ORR activity data were added in Table 2 and discussed (line 232-235).

Line 82: scheme 1: What is the duration of AL?

Authors’ response: Thanks. The duration of AL is 0.5 h, 1 h and 2 h, corresponding to the MA/EDTAFeNa-HT2(b), MA/EDTAFeNa-HT2(c) and MA/EDTAFeNa-HT2(d) in Table 2, respectively. The duration of AL was also specified in the caption of Figure.5 (line 264-265) and “3.2. Catalyst Preparation” (line 312-320). The duration of AL has been also added in Scheme 1.

Line 99: I would not call a standard lab-based XRD pattern wide angle XRD. To me wide angle diffraction can only be measured at the synchrotron.

Authors’ response: Thanks for the good suggestion.We have revised the unprecise description (Line 103)

Line 100/101: The used units look weird, please use: ° 2q (holds for the whole manuscript + SI)

Authors’ response: Thanks. They have been revised according to your guidance (Line 104-105)

Line 112: how is the saturated magnetization value determined? Where to find it in the graph?

Authors’ response: Thanks. The magnetization value is normalized to the mass of sample taken. The saturated magnetization value is defined as the magnetization value at the plateau (red circle).

Line 149: MA/EDTATeNa ??

Authors’ response: Thanks. The MA/EDTATeNa is not a sample; it means MA and EDTATeNa, which has been revised.

Line 168: Fig 3: I would show a y-axis as it might “calm” the figure.

Authors’ response: Thank you. A y-axis has been added in Figure 3.

Line 172: Fig 4: the denotation of the graphs is unfortunate. There is no a, b, c etc found in the figures.

Authors’ response: Thanks for friendly reminding. The denotations (a, b, c and d) have been supplemented in Figure 4A.

Line 159: What does „no N2 uptake“ mean?

Authors’ response: Thanks. “no N2 uptake” means the adsorbed N2 volume is nearly zero. We have corrected the expression (Line 168).

Line 181-184: How can you get from BET surface area to Fe particle size? The conclusion is far from clear to me.

Authors’ response: Thanks. In our experiments, the EDTAFeNa was pyrolized to obtain the EDTAFeNa-HT1, whose specific surface area and pore volume were very low. After the EDTAFeNa-HT1 was acid-leached, the as-prepared EDTAFeNa-AL had large specific surface area and pore volume, which indicate that the generation of mesopores should be attributed to removal of Fe species. Thus, it is concluded that the mesopore size matches with the particle size of Fe species. After the second heat-treatment, the resultant EDTAFeNa-HT2 had nearly the same specific surface area and pore size. When MA was introduced, the as-obtained MA/EDTAFeNa-HT2 possessed the same pore size to that of the EDTAFeNa-HT2. Based on the results, we concluded that particle size of Fe species was not influenced by introduction of MA. We have revised the expression (Line 191-195). If you think the conclusion is still far from clear to you, we will delete it. 

Sample

SBET

(m2 g-1)

V

(cm3 g-1)

Dp

(nm)

EDTAFeNa-HT1

17

0.03

4.0

EDTAFeNa-AL

225

0.41

3.9

EDTAFeNa-HT2

295

0.51

3.9

Line 186: How can one know that the generated Fe species were easily to be removed in this argumentation?

Authors’ response: We are sorry for a unclear statement. In our previous work (Int. J. Electrochem. Sci. 2019, 14, 6938), after the EDTAFeNa-HT1 was acid-leached with 0.5 M H2SO4 for 0.5 h, the saturation magnetization value of the as-prepared EDTAFeNa-AL was much lower than that for the EDTAFeNa-HT1, which indicates that most of the generated Fe species were easily to be removed. We have added the related statement (Line 191-195).  

Line 199: How do you know you have FeNx sites? There is no direct proof.

Authors’ response: Thanks. We really don’t know the presence of FeNx sites or not in the MA/EDTAFeNa-HT2 due to absence of some characterization techniques such as aberration-corrected scanning transmission electron microscopy, EXAFS and so on. We are sorry for that the presence of FeNx sites was only concluded based on references, removal of some crystalline Fe species and relatively higher ORR activity. We will strive to obtain above-mentioned characterization techniques for verifying the presence of FeNx sites in the following work.

Line 216: You cannot exclude the influence of Fe-Nx sites, because you have no proof or quantification for these sites.

Authors’ response: Thanks for your advice. We really have no proof or quantification for these sites. As well known, during the preparation of Fe−N-C catalysts, the generation of FeNxCy active sites is unavoidably accompanied with the stable metallic iron (Fe), iron carbide (Fe3C) and iron oxides (FexOy) particles due to the performed high-temperature pyrolysis step. It is perhaps concluded that the increase in numbers of active FeNx sites are mainly responsible for the higher ORR activity via increasing N content whilst nearly excluding effects of the graphitization degree, variety of crystalline iron species and mesoscopic structure. We have modified the unclear expressions (Line 237-240).

Line: I would say that Pt outperforms the Fe-N-C catalysts from this study, especially if you would determine the mass activities of each catalyst.

Authors’ response: Thanks. This is very insightful. We admitted that the MA/EDTAFeNa-HT2 is still inferior to the Pt/C. Actually, the catalyst loading is lower for the Pt/C than that for the MA/EDTAFeNa-HT2, as a result, the mass activity of the Pt/C outperforms that of the MA/EDTAFeNa-HT2 under the similar kinetic current densities. We have been optimizing the synthetic strategies for further improving the ORR activity of the Fe-N-C catalysts and have made some progress.

Line 223/224: Two consecutive one-electron steps? This is wrong.

Authors’ response: Thanks for pointing out the fault. We strongly agree with you. It has been verified and corrected (Line 248).

Line 226-228: This sentence is very general and has no distinct meaning or sense with respect to the presented data. Please specify and clarify.

Authors’ response: Thanks for good suggest. We have specified the ORR mechanism (Line 250-259).

Line 252: What potential was applied for the MeOH crossover measurements?

Authors’ response: Thank you. The MeOH crossover measurements were carried at at 0.85 V with rotation speed of 1600 rpm, which were described in “3.4. Electrochemical Measurements”(Line 348-351).

Line 255-257: What about the remaining currents after methanol crossover? How were these valued determined?

Authors’ response: Thanks. Before addition of methanol, the stable current density was about 2.56 mA cm-2. After methanol crossover, the current density decreased to about 2.03 mA cm-2 when it is at 6000 s with 79% of remaining current ( 2.03/2.56*100% = 79%).

Line 306/307: Loading in wrong unit? Rather µg/cm2

Authors’ response: Thanks for pointing out the fault. The 6 μg μL-1 is the concentration of catalyst ink. The catalyst loading should be 459 μg cm-2 ( 15 μL×6 μg μL-1/0.19625 cm2 = 459 μg cm-2), which has been corrected (Line 338).

Line 321: What is economical about this route? This point was not discussed in the manuscript at all.

Authors’ response: Thanks for good advice. We think this route is economical due to low-cost precursors and facile synthesis steps, which have been described in the revised manuscript (Line 65-67).

All changes were highlighted in yellow background.

Round 2

Reviewer 2 Report

In my opinion, the quality of the presented manuscript has clearly improved. That is why I would recommend the publication after the language has been further improved.